# Polyculture Affects the Growth, Antioxidant Status, Nutrient Content, and Flavor of Chinese Mitten Crabs (*Eriocheir sinensis*) and Largemouth Bass (*Micropterus salmoides*)

**Silu Che** [1] ⊙, **Shiheng Li** [1], **Quanjie Li** [2], **Yi Sun** [2], **Zhaowei Zheng** [3], **Zhijuan Nie** [2], **Zhonglin Tang** [4], **Peipei Wang** [4], **Jiancao Gao** [2] ⊙ and **Gangchun Xu** [1,2,3,*]

1 College of Fisheries and Life Science, Shanghai Ocean University, Shanghai 201306, China
2 Key Laboratory of Integrated Rice-Fish Farming Ecology, Ministry of Agriculture and Rural Affairs, Freshwater Fisheries Research Center, Chinese Academy of Fishery Sciences, Wuxi 214081, China
3 Wuxi Fisheries College, Nanjing Agricultural University, Wuxi 214081, China
4 Nanjing Fisheries Research Institute, Nanjing 210036, China
* Correspondence: xugc@ffrc.cn; Tel.: +86-0510-85390029

**Abstract:** Chinese mitten crabs (*Eriocheir sinensis*) and largemouth bass (*Micropterus salmoides*) are popular with consumers in China. In recent years, the polyculture of these two species has received more attention, but little is known about how their interactions affect their commercially important traits. In this study, we set up an *E. sinensis* monoculture group (EM), a *M. salmoides* monoculture group (MM), and a polyculture group containing both species (EP) and compared the growth parameters, antioxidant statuses, nutritional compositions, and flavor qualities of crabs and fish between the different culture modes. Growth parameters in male crabs and largemouth bass were significantly higher in the EP group than in the EM and MM groups, respectively. Crabs in the EM and EP groups did not differ significantly in malondialdehyde content or glutathione peroxidase activity, regardless of the sampling time, which suggests that crabs in these groups had similar antioxidant and immunity capacities. Compared to the MM group, the activities of superoxide dismutase and catalase of largemouth bass in the EP group were higher, indicating the superior antioxidant capacity of fish in the polyculture mode. Alkaline phosphatase and acid phosphatase activities of both crabs and largemouth bass fluctuated with time in all groups, indicating their important roles in maintaining the health of these cultured species. The amino acid and fatty acid contents of edible tissues were similar between the EM and EP groups and the MM and EP groups, suggesting comparable flavor and quality of edible tissues in crabs and largemouth bass between culture modes. This study provides theoretical support for the polyculture of Chinese mitten crabs and largemouth bass.

**Keywords:** Chinese mitten crab; largemouth bass; polyculture; growth parameters; flavor

## 1. Introduction

The Chinese mitten crab (*Eriocheir sinensis*) is an important aquaculture species that is widely distributed in eastern China [1–3]. Aquaculture of this species began in the 1980s, and large-scale production began in the late 1990s [4]. The production of *E. sinensis* reached 775,900 tons in 2020 according to the China Fishery Statistical Yearbook [5]. *E. sinensis* is a traditional savory food in China, and the cooked meat has a uniquely pleasant aroma and delicious taste. Therefore, it is very popular among Chinese consumers [6].

Largemouth bass (*Micropterus salmoides*) is native to North America and was introduced to China in the 1980s. Due to its delicious meat, high nutritional value, rapid growth, short breeding period, and strong adaptability, it quickly became one of the most important freshwater breeding varieties in China [7–11].

The growth, physiological statuses, and nutrient compositions of Chinese mitten crabs and largemouth bass are affected by many factors, including stocking density, feeding, and aquaculture method [12–15], and different farming techniques can improve the culture water environment. Because aquatic animals in an ideal polyculture pond occupy different ecological niches and have complementary feeding habits, farmers are often advised to develop polyculture systems to increase their productivity [16,17]. It is generally accepted that polyculture provides higher yields, and it is considered more ecologically healthy than monoculture [18,19]. Based on the common feeding habits and different habitats of Chinese mitten crabs and largemouth bass, the bass–crab polyculture model was developed. This polyculture model can significantly increase the output of aquaculture products, provide ecological benefits, efficiently and fully utilize space, form a mutually beneficial ecosystem, reduce morbidity, and improve the cultural environment [20].

The goal of this study was to compare the growth, physiological status, nutritional composition, and flavor quality of *E. sinensis* and *M. salmoides* between monoculture and polyculture modes. We measured the growth parameters and antioxidant activities at four time points. We also examined differences in nutritional value and flavor between the two culture modes by determining the proximate, fatty acid, and amino acid compositions of edible tissues using the samples collected at the last time point.

## 2. Materials and Methods

### 2.1. Ethics Statement

The experimental protocol was performed following the guidelines approved by the Institutional Animal Care and Use Committee of the Ministry of Freshwater Fisheries Research Center of the Chinese Academy of Fishery Sciences.

### 2.2. Experimental Design

The experiment was conducted at Yangzhong Experimental Base (Zhenjiang, China), Freshwater Fisheries Research Center, Chinese Academy of Fishery Sciences in January and November 2020. The Chinese mitten crabs used in the experiment came from Suzhou Youhua Ecological Technology Co., Ltd. (Suzhou, China). Largemouth bass were purchased from Anhui Zhanglin Fishery Co., Ltd. (Tongling, China)

Nine semi-enclosed ponds with convenient irrigation and drainage and consistent specifications were randomly selected. Each pond was (length × width × height = 75 m × 23 m × 1.5 m); microporous aeration pipes were installed at the bottoms of the ponds. We set up three groups in this study, each with three replicates: *E. sinensis* monoculture (EM), *M. salmoides* monoculture (MM), and polyculture of the two species (EP). The EM and MM were used as the respective control groups for the two species. Healthy and similar-sized crabs (12.53 ± 3.25 g) were stocked at 0.7 individuals per m$^2$ in six of the ponds. Three of these ponds were randomly selected and stocked with largemouth bass (184.62 ± 2.74 g) at a density of 2.2 individuals per m$^2$. The other three ponds were stocked only with largemouth bass at the same density.

Crabs were fed at 17:00 daily with a commercial diet (Table 1; Nanjing Aohua Biotechnology Co., Ltd., Nanjing, China). Largemouth bass were fed two times a day at 7:00 and 17:00, respectively, with a commercial diet (Table 1; Jiangsu Hipore Feed Co., Ltd., Xinghua, China). To detect water quality, we collected water samples every three days. During the rearing period, the dissolved oxygen of the pond was 8.24 ± 0.51 mg/L, the transparency was about 40 ± 10 cm, pH was maintained at 8.0 ± 0.5, ammonia nitrogen was 0.16 ± 0.03 mg/L, and nitrite nitrogen was 0.01 ± 0.01 mg/L.

### 2.3. Sampling Protocol

Samples of crabs were collected on 9 May (T1), 6 June (T2), 10 July (T3), and 10 October 2020 (T4). A total of 30 crabs (5 female crabs and 5 male crabs from each replicate pond) were randomly sampled on each date from the EM and EP groups. After being anesthetized on ice, the following measurements were taken: wet weight (using an electronic scale, accurate

to 0.01 g) and carapace length and width (using a digital caliper, accurate to 0.01 mm). Next, the hepatopancreas, gonad, and muscle from each crab were dissected carefully and weighed separately. These tissues were frozen in liquid nitrogen immediately and stored at –80 °C for later analysis. From the EM and EP groups, we collected an additional 18 crabs (3 female crabs and 3 male crabs from each of the three replicate ponds) from each time point for quality analysis of edible tissues.

**Table 1.** Formulation and proximate composition of the experimental diet (% dry weight).

| Composition | Crabs Feed | Largemouth Bass Feed |
|---|---|---|
| Moisture (%) | 12.02 ± 0.11 | 8.94 ± 2.15 |
| Crude protein (%) | 37.09 ± 2.33 | 51.32 ± 2.41 |
| Crude lipid (%) | 5.12 ± 0.18 | 8.65 ± 0.72 |
| Ash (%) | 10.66 ± 1.24 | 9.82 ± 0.62 |

Samples of the largemouth bass were collected on 9 May (T1), 6 June (T2), and 10 October 2020 (T4). Fish were fasted for 24 h before sampling. At each time point, a total of 45 fish (15 fish from each replicate pond) were caught randomly from the MM and EP groups and anesthetized with MS-222 (200 mg/L). Prior to dissection, each fish was weighed and measured to obtain body height, standard length, and body width. Blood samples were collected from the caudal vein using a hypodermic syringe. The blood samples were transferred to a centrifuge tube without anticoagulant and centrifuged to obtain serum samples. Hepatopancreas and muscle samples were also collected from those fish, frozen in liquid nitrogen, and stored at –80 °C for later analysis. From the MM and EP groups, we collected an additional 18 fish (6 from each of 3 replicate ponds) from each time point for quality analysis of edible tissues.

The gonadal somatic index (GSI) and hepatosomatic index (HSI) of the crabs were calculated as follows: GSI (%) = 100 × gonadal weight/body wet weight, and HSI (%) = 100 × hepatosomatic weight/body wet weight. The meat yield (MY) of the crabs was calculated as MY (%) = 100 × muscle wet weight/body wet weight, and total edible yield (TEY, %) was calculated as the sum of GSI, HSI, and MY. The condition factor (CF[$g/cm^3$]) was calculated as the body wet weight/carapace length$^3$ [21].

The relevant parameters of the largemouth bass were calculated as follows: HSI (%) = 100 × hepatopancreas weight/final body weight; viscerosomatic index (VSI, %) = 100 × viscera weight/body weight; weight gain (WG, %) = $100 \times (W_t - W_0)/W_0$; and (CF[$g/cm^3$]) = body weight/body length$^3$. $W_t$ is the final body weight (g), and $W_0$ is the initial body weight (g) [22].

### 2.4. Enzyme Activity Assays

To evaluate the physiological status of the crabs and fish, we measured enzyme activity in the hepatopancreas. We used ice-cold physiological saline at 0.9 mL saline per 0.1 g tissue to treat the tissue samples and then centrifuged them at 2500 rpm for 10 min. The supernatant was collected to analyze the content of malondialdehyde (MDA; A003-1), the activities of glutathione peroxidase (GSH-Px; A005), superoxide dismutase (SOD; A001-3) and catalase (CAT; A007-1), acid phosphatase (ACP; A060-2), and alkaline phosphatase (AKP; A059-2) using kits and following the manufacturer's instructions [23]. We used Coomassie brilliant blue staining assay kits (A045- 2-2) to quantify total protein. All kits were purchased from Nanjing Jiancheng Biology Engineering Institute (Nanjing, Jiangsu, China).

### 2.5. Biochemical Analysis

To detect differences in the edible tissues between culture modes, we compared the proximate composition and fatty acid composition of crab and largemouth bass edible tissues and the amino acid composition of the muscle of largemouth bass (EM vs. EP and MM vs. EP). Moisture, ash, total lipid, and crude protein contents of edible tissues were

measured using standard methods. Briefly, these tissues were dried to constant weights at 105 °C to determine their moisture content [24], while the crude protein content was determined using the Kjeldahl method (using a 6.25 N to protein conversion factor). The ash content was determined using a muffle furnace at 550 °C to a constant weight [25], while the total lipid content was measured by Soxhlet extraction with petroleum ether [26].

Fatty acid composition, amino acid composition, and flavor-enhancing nucleotides of edible tissues were evaluated at the National Key Laboratory of Food Science and Technology, Jiangnan University (Wuxi, Jiangsu province, China) [27]. For fatty acid analysis, fatty acid methyl esters were prepared by transesterification with 2 M potassium hydroxide in methanol for 20 min. After centrifugation for 5 min at 805× g, the upper organic layer was diluted with hexane and used for gas chromatography analysis. Fatty acid methyl esters were analyzed using an Agilent 7820A gas chromatograph (Agilent, Santa Clara, CA, USA) equipped with a hydrogen flame ionization detector [28]. The fatty acid composition was expressed as a percentage of each fatty acid to the total fatty acids (%). After hydrolysis by 6 M hydrochloric acid, the amino acid composition of muscle samples was determined following the general rules for amino acid analysis (JY/T019-1996) using an Agilent 1100 HPLC system [29].

### 2.6. Statistical Analysis

Normal distribution and homogeneity of variance of data were tested using Shapiro–Wilk and Levene tests ($\alpha = 0.05$), respectively. When necessary, logarithmic transformation was performed before data analysis. For enzyme activity and fatty acid and nutrient contents, we used independent sample $t$-tests to test for potential differences between the monoculture and polyculture modes. $p < 0.05$ was considered statistically significant. All statistical analyses were performed using IBM SPSS Statistics 26.0 (IBM Inc., Armonk, NY, USA).

## 3. Results

### 3.1. Growth Performance

As shown in Table 2, the growth parameters of female crabs in the EP group were slightly higher than those in the EM group, but the differences were not statistically significant ($p > 0.05$). In contrast, the body weight, carapace length, carapace width, and carapace height of male crabs in the EP group were significantly higher than those in the EM group ($p < 0.05$). The MY and CF of male crabs in the EP group were higher than those of female crabs in the EP group ($p > 0.05$).

**Table 2.** Growth parameters of Chinese mitten crabs reared in the EM and EP groups from T1 to T4. Values are means $\pm$ SD ($n = 30$).

| Growth Parameters | Female Crabs | | Male Crabs | |
|---|---|---|---|---|
| | EM | EP | EM | EP |
| Body weight (g) | 173.86 ± 32.89 | 204.57 ± 33.07 | 217.09 ± 29.57 | 283.21 ± 17.00 * |
| Carapace length (mm) | 70.46 ± 4.72 | 74.14 ± 3.10 | 72.95 ± 2.26 | 77.76 ± 1.24 * |
| Carapace width (mm) | 74.78 ± 4.76 | 77.70 ± 3.96 | 78.41 ± 2.38 | 83.98 ± 2.02 * |
| Carapace height (mm) | 38.48 ± 3.08 | 41.91 ± 2.62 | 38.29 ± 0.65 | 40.66 ± 1.67 * |
| Gonadosomatic index (%) | 5.41 ± 2.73 | 5.65 ± 0.82 | 1.26 ± 0.41 | 1.44 ± 0.33 |
| Hepatosomatic index (%) | 10.15 ± 1.29 | 10.31 ± 1.62 | 8.44 ± 0.58 | 8.89 ± 1.63 |
| Meat yield (%) | 18.97 ± 1.84 | 18.05 ± 4.68 | 18.00 ± 2.00 | 18.83 ± 2.85 |
| Total edible yield (%) | 34.53 ± 3.33 | 34.01 ± 5.57 | 27.70 ± 2.49 | 29.16 ± 4.17 |
| Condition factor × 100 (g/cm$^3$) | 49.31 ± 2.84 | 49.89 ± 3.58 | 55.75 ± 4.75 | 60.22 ± 2.78 |

* Data with asterisks in the same row are significantly different in the EM vs. EP comparison ($p < 0.05$).

The final body weight and weight gain rate of largemouth bass in the EP group were significantly higher than those in the MM group, and the HSI was significantly lower than

that in the MM group ($p < 0.05$). There was no significant difference in CF and VSI ($p > 0.05$) between the MM and EP groups (Table 3).

**Table 3.** Growth parameters of largemouth bass reared in the MM and EP groups from T1 to T3. Values are means $\pm$ SD ($n = 45$).

| Growth Parameters | Largemouth Bass | |
|---|---|---|
| | MM | EP |
| Final body weight (g) | $560.51 \pm 40.70$ | $632.29 \pm 35.01$ * |
| Weight gain (%) | $204.63 \pm 22.12$ | $243.63 \pm 19.03$ * |
| Condition factor $\times$ 100 (g/cm$^3$) | $2.54 \pm 0.26$ | $2.53 \pm 0.13$ |
| Hepatosomatic index (%) | $2.66 \pm 0.44$ * | $1.93 \pm 0.45$ |
| Viscerosomatic index (%) | $7.95 \pm 1.02$ | $7.33 \pm 0.88$ |

* Data with asterisks in the same row are significantly different ($p < 0.05$).

*3.2. Antioxidant Status*

The MDA content and GSH-Px activity of female and male crabs at different developmental stages did not differ significantly between the EM and EP groups ($p > 0.05$ Figure 1a–d). During the T1 period, the ACP activity of female crabs in the EM group was significantly higher than that in the EP group ($p < 0.05$, Figure 2a), but there was no significant difference in AKP activity between the two groups ($p > 0.05$, Figure 2c). At T4, the AKP activity of female crabs in the EP group was significantly higher than that at the other stages ($p < 0.05$, Figure 2c). There was no significant difference in the activity of ACP and AKP of male crabs between the EM and EP groups ($p > 0.05$, Figure 2b,d). In the EP group, the ACP activity of male crabs showed an upward trend over time (Figure 2b). ACP and AKP activity of male crabs was highest at T4. ACP and AKP activities of male crabs differed significantly at different sampling times in the EP group ($p < 0.05$, Figure 2b,d).

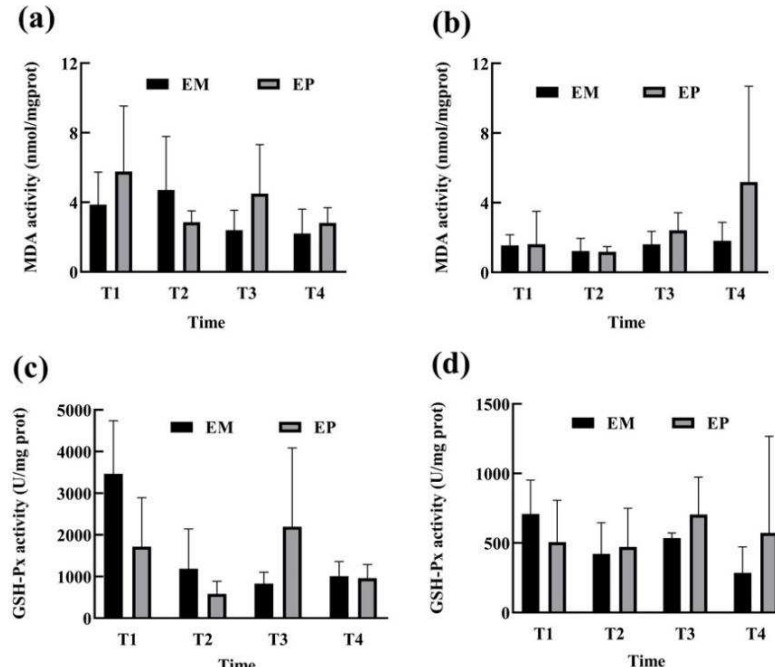

**Figure 1.** Differences in MDA content in (**a**) female and (**b**) male crabs and differences in GSH-Px activity in (**c**) female and (**d**) male crabs between the EM and EP groups at different time points. Data are means $\pm$ SD.

The CAT activity of largemouth bass did not differ significantly between the two groups at T1, but it was significantly higher in the EP group than in the MM group at T2 (Figure 3a). The GSH-Px activity in the EP group was significantly lower than that in the MM group

(Figure 3b), and the SOD activity was significantly higher in the EP group than in the MM group (Figure 3c) at both T1 and T2. The ACP activity in the EP group was significantly higher than that in the MM group at T1 ($p < 0.05$), but there was no significant difference between the two groups at T2 ($p > 0.05$) (Figure 3d). The activity of AKP in the EP group was significantly higher than that in the MM group at both time points ($p < 0.05$) (Figure 3e).

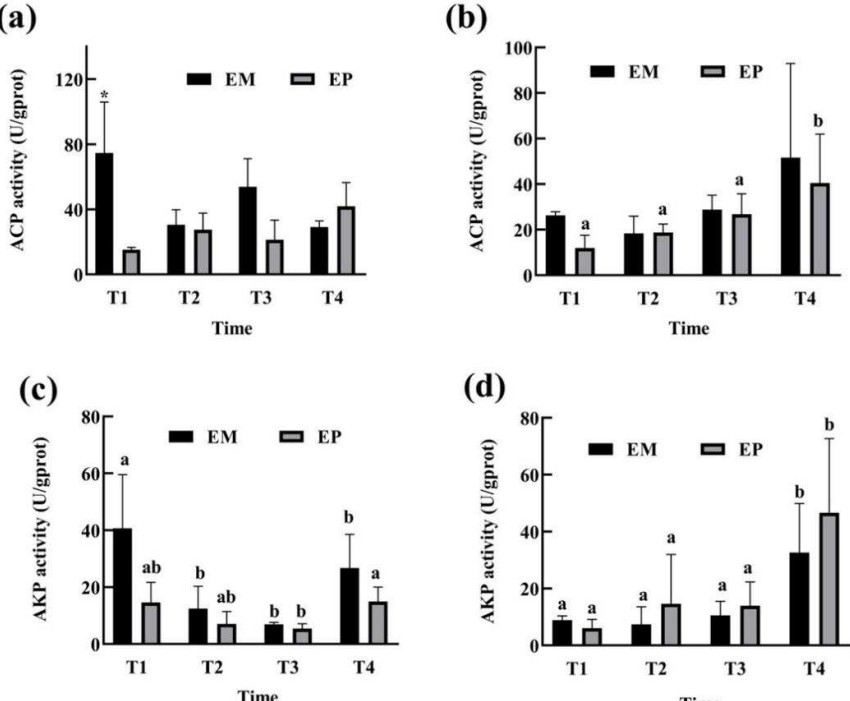

**Figure 2.** Differences in activities of ACP in (**a**) female and (**b**) male crabs and differences in AKP activity in (**c**) female and (**d**) male crabs between the EM and EP groups at different time points. Data are means ± SD. * indicates a statistically significant difference between the EM and EP groups at T1. Different lowercase letters indicate significant differences among different time points in the same group ($p < 0.05$).

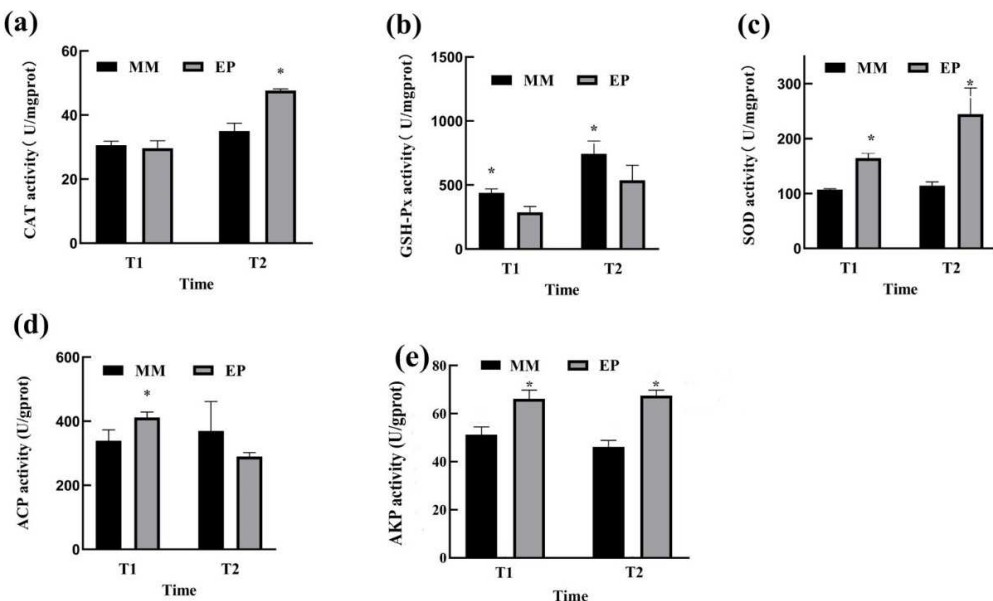

**Figure 3.** Differences in (**a**) CAT, (**b**) GSH-Px, (**c**) SOD, (**d**) ACP, and (**e**) AKP activities in largemouth bass at different time points. Data are means ± SD. * indicates statistically significant differences between the MM and EP groups at each time point ($p < 0.05$).

### 3.3. Proximate Compositions of Edible Tissues

Figure 4 shows the proximate composition of edible tissues of *E. sinensis*. Moisture and crude fat contents were higher than ash and crude protein contents, and no gender difference was detected. Additionally, no significant differences in the conventional nutritional components were found between crabs in the EM and EP groups ($p > 0.05$).

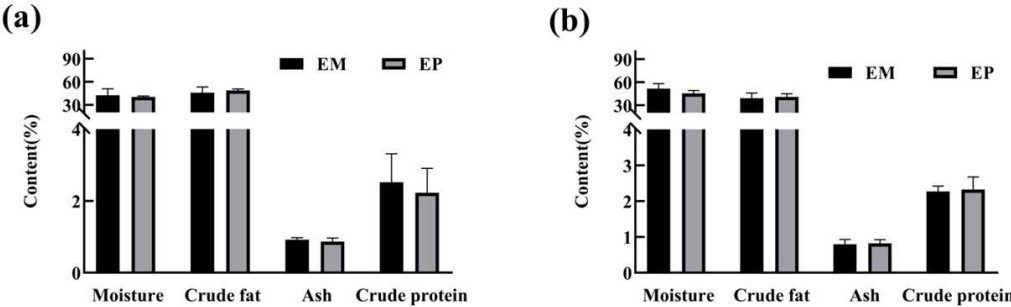

**Figure 4.** Proximate composition of edible tissues of Chinese mitten crabs in the EM and EP groups: (**a**) female; (**b**) male.

In largemouth bass muscle tissue, the ash content in the EP group was significantly higher than that in the MM group ($p < 0.05$) (Figure 5), but no significant differences in moisture, crude protein, and crude fat contents were detected between the two groups ($p > 0.05$).

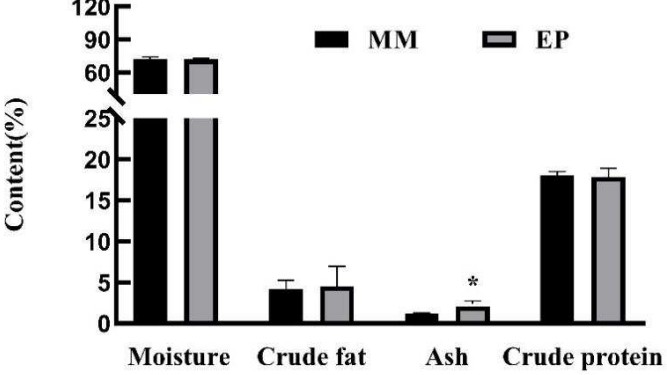

**Figure 5.** Proximate composition of largemouth bass muscle in the EM and EP groups. * indicates the statistically significant difference in the EM and EP groups ($p < 0.05$).

### 3.4. Composition of Fatty Acids in Edible Tissues

Twenty-one kinds of fatty acids were detected in the edible tissues of crabs and largemouth bass, including eight kinds of saturated fatty acids (SFAs), four kinds of monounsaturated fatty acids (MUFAs), and nine kinds of polyunsaturated fatty acid (PUFAs; Table 4).

In male crabs, the content of SFA C17:0 was significantly lower and that of C18:0 was significantly higher in the EM group compared to the EP group ($p < 0.05$). The contents of MUFAs C16:1 and C18:1 were significantly lower and higher, respectively, in the EM group than in the EP group ($p < 0.05$). The most abundant PUFA in both groups was C18:2. The content of C18:3 was significantly lower and that of C22:5 was significantly higher in the EM group compared to the EP group ($p < 0.05$).

In largemouth bass, C18:2 and C20:2 contents were significantly higher in the EP group than in the MM group ($p < 0.05$), whereas the content of C14:0 was higher in the MM group ($p < 0.05$). Significant differences in other fatty acid contents between the two groups were not detected ($p > 0.05$).

**Table 4.** Fatty acid composition (% total fatty acids) of edible tissues in Chinese mitten crabs and largemouth bass in the EM, MM, and EP groups. Values are means $\pm$ SD ($n$ = 18).

| Fatty Acids | Female Crabs | | Male Crabs | | Largemouth Bass | |
|---|---|---|---|---|---|---|
| | EM | EP | EM | EP | MM | EP |
| C12:0 | 0.13 ± 0.03 | 0.17 ± 0.01 | 0.11 ± 0.04 | 0.18 ± 0.06 | 0.04 ± 0.02 | 0.02 ± 0.00 |
| C14:0 | 1.50 ± 0.09 | 1.57 ± 0.18 | 1.47 ± 0.04 | 1.98 ± 0.77 | 1.32 ± 0.07 * | 1.19 ± 0.06 |
| C15:0 | 0.32 ± 0.05 | 0.27 ± 0.02 | 0.32 ± 0.06 | 0.33 ± 0.03 | 0.20 ± 0.01 | 0.20 ± 0.01 |
| C16:0 | 21.77 ± 0.24 | 23.11 ± 1.39 | 21.85 ± 1.54 | 22.58 ± 0.24 | 18.68 ± 0.42 | 18.66 ± 0.61 |
| C16:1 | 12.68 ± 0.78 | 12.15 ± 0.71 | 9.88 ± 2.48 | 14.91 ± 0.48 * | 3.13 ± 0.20 | 3.04 ± 0.23 |
| C17:0 | 0.27 ± 0.01 | 1.61 ± 0.28 * | 0.32 ± 0.03 | 1.36 ± 0.16 * | 0.20 ± 0.01 | 0.20 ± 0.01 |
| C18:0 | 2.87 ± 0.18 | 3.09 ± 0.49 | 3.15 ± 0.35 * | 2.34 ± 0.30 | 6.85 ± 0.05 | 6.64 ± 0.30 |
| C18:1 | 34.93 ± 0.13 | 32.70 ± 0.23 | 34.87 ± 1.03 * | 29.78 ± 1.70 | 20.74 ± 0.78 | 20.36 ± 0.72 |
| C18:2 | 15.04 ± 1.47 | 14.51 ± 1.52 | 17.53 ± 3.26 | 15.96 ± 1.15 | 22.64 ± 0.35 | 23.83 ± 0.08 * |
| C18:3 | 2.48 ± 0.34 | 3.81 ± 0.50 * | 2.25 ± 0.64 | 3.61 ± 0.15 * | 1.52 ± 0.03 | 1.63 ± 0.11 |
| C20:0 | 0.16 ± 0.03 | 0.17 ± 0.02 | 0.11 ± 0.04 | 0.18 ± 0.10 | 0.23 ± 0.05 | 0.18 ± 0.03 |
| C20:1 | 1.45 ± 0.41 | 1.52 ± 0.22 | 1.43 ± 0.15 | 1.21 ± 0.28 | 0.82 ± 0.07 | 0.88 ± 0.13 |
| C20:2 | 1.00 ± 0.12 | 1.14 ± 0.06 | 0.98 ± 0.21 | 0.81 ± 0.16 | 0.80 ± 0.07 | 0.93 ± 0.06 * |
| C20:3 | 0.28 ± 0.02 * | 0.14 ± 0.01 | 0.36 ± 0.17 | 0.14 ± 0.01 | 0.23 ± 0.04 | 0.28 ± 0.07 |
| C20:4 | 1.12 ± 0.30 | 1.18 ± 0.09 | 1.09 ± 0.07 | 1.34 ± 0.15 | 1.65 ± 0.12 | 1.58 ± 0.08 |
| C20:5 | 0.98 ± 0.27 | 0.92 ± 0.15 | 1.08 ± 0.55 | 1.26 ± 0.18 | 2.26 ± 0.14 | 2.01 ± 0.18 |
| C22:0 | 0.05 ± 0.01 | 0.05 ± 0.01 | 0.05 ± 0.00 | 0.05 ± 0.02 | 0.17 ± 0.06 | 0.19 ± 0.04 |
| C22:1 | 0.10 ± 0.03 | 0.07 ± 0.02 | 0.08 ± 0.03 | 0.08 ± 0.03 | 0.17 ± 0.04 | 0.30 ± 0.08 |
| C22:4 | 0.25 ± 0.02 | 0.17 ± 0.18 | 0.28 ± 0.15 | 0.04 ± 0.01 | 0.52 ± 0.04 | 0.56 ± 0.04 |
| C22:5 | 0.38 ± 0.05 * | 0.15 ± 0.04 | 0.47 ± 0.10 * | 0.14 ± 0.02 | 1.68 ± 0.20 | 1.49 ± 0.15 |
| C22:6 | 1.24 ± 0.16 * | 0.53 ± 0.26 | 1.45 ± 0.39 | 0.58 ± 0.07 | 16.15 ± 0.80 | 15.85 ± 0.69 |
| ∑PUFA | 22.78 ± 0.34 | 22.56 ± 0.94 | 25.49 ± 2.89 | 23.88 ± 1.22 | 47.45 ± 0.66 | 48.16 ± 1.20 |
| ∑SFA | 27.06 ± 0.28 | 30.05 ± 1.47 * | 27.38 ± 1.30 | 29.01 ± 0.61 | 27.70 ± 0.31 | 27.28 ± 0.36 |
| ∑PUFA/∑SFA | 0.84 ± 0.01 | 0.75 ± 0.07 | 0.94 ± 0.15 | 0.82 ± 0.04 | 1.71 ± 0.01 | 1.77 ± 0.07 |

* Data with asterisks in the same row for a given comparison are significantly different ($p < 0.05$). In the comparison of SFAs in female crabs between the EM and EP groups, the content of C17:0 and total saturated fatty acids (∑SFA) was significantly higher ($p < 0.05$) in the EP group, but the content of the other SFAs did not differ significantly ($p > 0.05$). Among the PUFAs, the content of C18:2 was highest in both groups, the content of C18:3 was significantly lower, and those of C20:3, C22:5, and C22:6 were significantly higher in the EM group ($p < 0.05$).

### 3.5. Amino Acid Composition

The most abundant hydrolyzed amino acid in the muscle of crabs was glutamate (Glu), followed by leucine (Leu) and aspartic acid (Asp) (Table 5). The essential amino acid (EAA) ratio (E/T) of female crabs in both the EM and EP groups was 39%, and in female crabs, no significant difference in the content of hydrolyzed amino acids was detected between the two groups ($p > 0.05$). However, the E/T of male crabs in the EM and EP groups was 38% and 39%, respectively, and the difference was statistically significant ($p < 0.05$). The non-essential amino acid (NEAA) ratio (N/T) also differed significantly between the EM and EP groups ($p < 0.05$). In largemouth bass, no significant differences in hydrolyzed amino acid content, total EEAs, and total NEAAs were detected between the MM and EP groups ($p > 0.05$).

Seventeen free amino acids (FAAs) were detected in the tissues analyzed in this experiment, including 10 NEAAs and 7 EAAs (Table 6). Asp, histidine (His), serine (Ser), arginine (Arg), proline (Pro), tyrosine (Tyr), threonine (Thr), valine (Val), phenylalanine (Phe), methionine (Met), isoleucine (Ile), lysine (Lys), Leu, and total FAAs (T) contents and the E/T were significantly higher in female than in male crabs in the EP group. In female crabs, the proportion of EEAs in the EM and EP groups was 39% and 38%, respectively. In male crabs, the contents of Arg, Pro, and alanine (Ala) were highest. The content of Arg of the female crabs in the EP group was significantly higher than that in the EM group ($p < 0.05$), but no significant differences were detected for the other FAAs ($p > 0.05$). The proportion of EEAs in male crabs was 32% in both the EM and EP groups. The main FAAs in both male and female crabs were Arg and Ala. In largemouth bass, the contents of Asp, Glu, Val, Met, and Ile in the MM group were significantly higher than those in the EP group

($p < 0.05$), but there was no significant difference in the other FAAs or total amino acids between the two groups ($p > 0.05$).

**Table 5.** Hydrolyzed amino acids (g/100 g tissue) detected in the muscles of Chinese mitten crabs and largemouth bass cultured in the EM, MM, and EP groups. Values are means $\pm$ SD ($n$ = 18).

| Hydrolyzed Amino Acids | Female Crabs | | Male Crabs | | Largemouth Bass | |
|---|---|---|---|---|---|---|
| | EM | EP | EM | EP | MM | EP |
| Asp | 1.70 ± 0.17 | 1.66 ± 0.14 | 1.48 ± 0.26 | 1.36 ± 0.10 | 8.44 ± 0.41 | 8.44 ± 0.20 |
| Glu | 2.24 ± 0.23 | 2.22 ± 0.18 | 2.06 ± 0.34 | 1.87 ± 0.13 | 13.15 ± 0.58 | 13.20 ± 0.25 |
| Ser | 0.62 ± 0.04 | 0.67 ± 0.07 | 0.57 ± 0.10 | 0.53 ± 0.06 | 2.65 ± 0.06 | 2.69 ± 0.04 |
| His | 0.49 ± 0.04 | 0.49 ± 0.03 | 0.41 ± 0.07 | 0.41 ± 0.03 | 2.23 ± 0.11 | 2.25 ± 0.06 |
| Gly | 0.85 ± 0.07 | 0.86 ± 0.08 | 0.76 ± 0.11 | 0.71 ± 0.11 | 3.95 ± 0.22 | 4.04 ± 0.18 |
| Arg | 1.05 ± 0.18 | 1.10 ± 0.06 | 1.08 ± 0.28 | 0.82 ± 0.10 | 4.65 ± 0.22 | 4.60 ± 0.07 |
| Ala | 0.97 ± 0.11 | 1.00 ± 0.06 | 0.97 ± 0.21 | 0.85 ± 0.06 | 4.74 ± 0.21 | 4.73 ± 0.08 |
| Tyr | 0.57 ± 0.14 | 0.45 ± 0.04 | 0.62 ± 0.25 | 0.38 ± 0.08 | 2.48 ± 0.13 | 2.77 ± 0.52 |
| Pro | 0.78 ± 0.13 | 0.76 ± 0.32 | 0.78 ± 0.29 | 0.63 ± 0.03 | 2.56 ± 0.23 | 2.32 ± 0.18 |
| Cys-s | 0.08 ± 0.00 | 0.09 ± 0.01 | 0.07 ± 0.07 | 0.06 ± 0.01 | 0.37 ± 0.08 | 0.40 ± 0.01 |
| Thr | 0.86 ± 0.07 | 0.89 ± 0.09 | 0.77 ± 0.11 | 0.72 ± 0.07 | 3.27 ± 0.13 | 3.26 ± 0.04 |
| Val | 0.96 ± 0.12 | 0.99 ± 0.07 | 0.92 ± 0.19 | 0.76 ± 0.06 | 4.34 ± 0.24 | 4.34 ± 0.11 |
| Met | 0.48 ± 0.12 | 0.37 ± 0.09 | 0.39 ± 0.11 | 0.30 ± 0.09 | 2.48 ± 0.11 | 2.46 ± 0.05 |
| Phe | 0.81 ± 0.10 | 0.79 ± 0.06 | 0.71 ± 0.15 | 0.62 ± 0.06 | 3.42 ± 0.16 | 3.43 ± 0.06 |
| Ile | 0.75 ± 0.10 | 0.74 ± 0.05 | 0.67 ± 0.13 | 0.58 ± 0.05 | 4.10 ± 0.21 | 4.09 ± 0.09 |
| Leu | 1.21 ± 0.16 | 1.27 ± 0.09 | 1.19 ± 0.19 | 0.99 ± 0.07 | 6.39 ± 0.29 | 6.42 ± 0.13 |
| Lys | 1.00 ± 0.16 | 0.95 ± 0.19 | 0.99 ± 0.21 | 0.80 ± 0.07 | 8.37 ± 0.36 | 8.47 ± 0.16 |
| E | 6.07 ± 0.83 | 6.01 ± 0.52 | 5.64 ± 1.09 | 4.77 ± 0.45 | 32.38 ± 1.48 | 32.48 ± 0.65 |
| N | 9.33 ± 0.97 | 9.28 ± 0.67 | 8.81 ± 1.82 | 7.64 ± 0.66 | 45.24 ± 1.95 | 45.44 ± 1.17 |
| T | 15.40 ± 1.80 | 15.29 ± 1.19 | 14.45 ± 2.90 | 12.41 ± 1.11 | 77.62 ± 3.43 | 77.93 ± 1.73 |
| E/T | 0.39 ± 0.01 | 0.39 ± 0.00 | 0.39 ± 0.00 * | 0.38 ± 0.00 | 0.42 ± 0.00 | 0.42 ± 0.00 |
| N/T | 0.61 ± 0.01 | 0.61 ± 0.00 | 0.61 ± 0.00 | 0.62 ± 0.00 * | 0.58 ± 0.00 | 0.58 ± 0.00 |

* Data with asterisks in the same row for a given comparison are significantly different ($p < 0.05$).

**Table 6.** FAAs (g/100 g tissue) detected in the muscles of Chinese mitten crabs and largemouth bass cultured in the EM, MM, and EP groups. Values are means $\pm$ SD ($n$ = 18).

| Free Amino Acids | Female Crabs | | Male Crabs | | Largemouth Bass | |
|---|---|---|---|---|---|---|
| | EM | EP | EM | EP | MM | EP |
| Asp | 0.05 ± 0.01 | 0.06 ± 0.00 | 0.04 ± 0.02 | 0.05 ± 0.02 | 0.17 ± 0.01 * | 0.16 ± 0.00 |
| Glu | 0.17 ± 0.02 | 0.15 ± 0.02 | 0.19 ± 0.04 | 0.19 ± 0.04 | 0.78 ± 0.14 * | 0.20 ± 0.03 |
| Ser | 0.05 ± 0.00 | 0.05 ± 0.01 | 0.04 ± 0.02 | 0.04 ± 0.02 | 0.06 ± 0.01 | 0.06 ± 0.01 |
| His | 0.05 ± 0.00 | 0.05 ± 0.00 | 0.04 ± 0.02 | 0.03 ± 0.02 | 2.88 ± 0.25 | 3.02 ± 0.38 |
| Gly | 0.13 ± 0.01 | 0.12 ± 0.02 | 0.13 ± 0.02 | 0.13 ± 0.06 | 2.35 ± 0.42 | 2.63 ± 0.27 |
| Arg | 0.30 ± 0.04 | 0.38 ± 0.02 * | 0.30 ± 0.09 | 0.26 ± 0.11 | 0.07 ± 0.00 | 0.08 ± 0.01 |
| Ala | 0.23 ± 0.03 | 0.23 ± 0.00 | 0.32 ± 0.11 | 0.25 ± 0.02 | 1.14 ± 0.21 | 0.89 ± 0.04 |
| Tyr | 0.10 ± 0.02 | 0.11 ± 0.01 | 0.09 ± 0.04 | 0.07 ± 0.04 | 0.08 ± 0.00 | 0.10 ± 0.02 |
| Cys-s | 0.02 ± 0.00 | 0.02 ± 0.00 | 0.02 ± 0.00 | 0.02 ± 0.00 | 0.01 ± 0.00 | 0.01 ± 0.00 |
| Pro | 0.21 ± 0.06 | 0.19 ± 0.02 | 0.22 ± 0.17 | 0.17 ± 0.04 | 2.50 ± 1.10 | 1.65 ± 0.40 |
| Thr | 0.12 ± 0.00 | 0.10 ± 0.01 | 0.09 ± 0.04 | 0.08 ± 0.03 | 1.01 ± 0.05 | 0.86 ± 0.04 |
| Val | 0.12 ± 0.00 | 0.11 ± 0.01 | 0.09 ± 0.05 | 0.08 ± 0.04 | 0.15 ± 0.03 * | 0.11 ± 0.01 |
| Met | 0.06 ± 0.00 | 0.06 ± 0.00 | 0.06 ± 0.03 | 0.04 ± 0.02 | 0.06 ± 0.00 * | 0.04 ± 0.00 |
| Phe | 0.10 ± 0.02 | 0.10 ± 0.01 | 0.08 ± 0.05 | 0.07 ± 0.04 | 0.09 ± 0.01 | 0.08 ± 0.01 |
| Ile | 0.09 ± 0.00 | 0.09 ± 0.01 | 0.07 ± 0.04 | 0.06 ± 0.04 | 0.10 ± 0.02 * | 0.06 ± 0.01 |
| Leu | 0.18 ± 0.03 | 0.18 ± 0.02 | 0.15 ± 0.09 | 0.13 ± 0.07 | 0.20 ± 0.04 | 0.13 ± 0.01 |
| Lys | 0.17 ± 0.04 | 0.20 ± 0.02 | 0.16 ± 0.07 | 0.14 ± 0.08 | 0.59 ± 0.18 | 0.87 ± 0.32 |
| E | 0.84 ± 0.10 | 0.85 ± 0.07 | 0.70 ± 0.36 | 0.61 ± 0.32 | 2.20 ± 0.29 | 2.15 ± 0.35 |
| N | 1.32 ± 0.16 | 1.36 ± 0.05 | 1.39 ± 0.40 | 1.19 ± 0.32 | 10.09 ± 1.26 | 8.78 ± 0.20 |
| T | 2.16 ± 0.26 | 2.20 ± 0.12 | 2.10 ± 0.76 | 1.80 ± 0.65 | 12.29 ± 1.46 | 10.39 ± 0.35 |

**Table 6.** *Cont.*

| Free Amino Acids | Female Crabs | | Male Crabs | | Largemouth Bass | |
|---|---|---|---|---|---|---|
| | EM | EP | EM | EP | MM | EP |
| E/T | 0.39 ± 0.00 | 0.38 ± 0.01 | 0.32 ± 0.05 | 0.32 ± 0.06 | 0.18 ± 0.02 | 0.20 ± 0.03 |
| N/T | 0.61 ± 0.00 | 0.62 ± 0.01 | 0.68 ± 0.05 | 0.68 ± 0.06 | 0.82 ± 0.02 | 0.80 ± 0.03 |
| E/N | 0.64 ± 0.00 | 0.63 ± 0.03 | 0.48 ± 0.11 | 0.49 ± 0.13 | 0.22 ± 0.03 | 0.25 ± 0.04 |

* Data with asterisks in the same row for a given comparison are significantly different ($p < 0.05$).

### 3.6. Comparison of Nucleotide Content in Edible Tissues

Figure 6 shows the differences in nucleotide content in the edible tissues between crabs in the EM and EP groups. In both groups, adenosine (AMP), followed by inosine (HxR), were the most abundant flavorful nucleotides in crabs, and female crabs had higher levels of AMP than male crabs. The contents of inosinic acid (IMP) and HxR of female crabs in the EP group were slightly higher than those in the EM group, but the differences were not statistically significant ($p > 0.05$). The AMP content of male crabs in the EP group was slightly lower and HxR content was slightly higher than those in the EM group, but the differences were not statistically significant ($p > 0.05$).

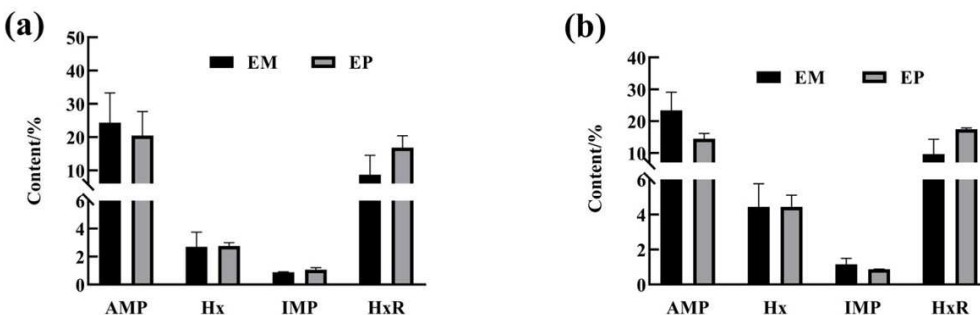

**Figure 6.** Comparison of nucleotide content in edible tissues of Chinese mitten crabs in the EM and EP groups: (**a**) female; (**b**) male.

The differences in nucleotide content in largemouth bass muscle between fish in the MM and EP groups are shown in Figure 7. Of the five nucleotides detected (IMP, GMP, UMP, CMP, and AMP), IMP was the most dominant and CMP was the least abundant. The GMP content in the EP group was significantly higher than that in the MM group ($p < 0.05$), but no significant differences between the MM and EP groups were detected for the other nucleotides ($p > 0.05$).

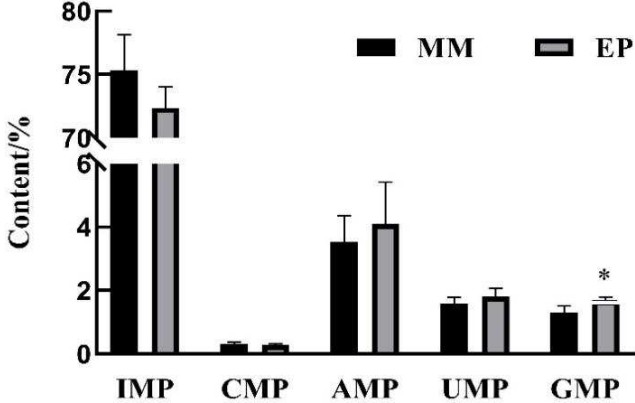

**Figure 7.** Comparison of nucleotide content in largemouth bass muscle in the EM and EP groups. * indicates the statistically significant difference in the EM and EP groups ($p < 0.05$).

## 4. Discussion

In recent years, increased pond stocking density, relatively simple culture mode structure, and excess feed have resulted in the deterioration of the aquaculture water environment and the frequent occurrence of diseases, which have negatively affected the development of the aquaculture industry. Therefore, culture mode in aquaculture has become an urgent issue in the aquaculture industry. The purpose of this study was to develop a mutually beneficial ecosystem for Chinese mitten crabs and largemouth bass culture using the green and healthy polyculture mode and to evaluate the potential benefits of this culture mode on crab and fish production.

We did not detect a significant difference in the growth parameters of female crabs between the monoculture and polyculture modes, but the growth parameters in male crabs were significantly higher in the EP group compared with the EM group. The final body weight and weight gain rate of largemouth bass in the EP group were significantly higher than those in the EM group. Costa et al. previously reported that the polyculture of shrimp and mullet reared together in earthen ponds favors the production of mullets [30]. Generally, many benefits have been achieved in aquatic polyculture systems when using fish, bivalves, and seaweeds, including improved growth [31]. The results of our study show that the polyculture mode can improve the specifications of commercial Chinese mitten crabs, especially male crabs, and those of largemouth bass size.

Similar to higher vertebrates, fish are susceptible to the effects of reactive oxygen species and have innate and effective antioxidant defenses [32]. Crustaceans have no adaptive immune system, so their antioxidant defense systems play an important role in coping with environmental stress. The antioxidant defense system of crustaceans consists of enzymatic and non-enzymatic antioxidant systems that protect organisms against oxidative stress and damage. SOD, CAT, and GSH-Px are important antioxidants [33,34]. SOD is a key antioxidant that can protect living cells from the oxidative damage of superoxide radicals by catalyzing the conversion of superoxide anion radicals ($O^{2-}$) into $H_2O_2$ and $O_2$. GSH-Px is one of the main protectors against lipid peroxidation damage, and it promotes the decomposition of $H_2O_2$ and reduces toxic peroxides to non-toxic hydroxyl compounds. CAT can also convert $H_2O_2$ to water in order to thoroughly clear oxyradicals [35,36]. MDA, the final breakdown product of lipid peroxides, has strong cytotoxicity, thus its content is often used to assess the degree of oxidative damage to cells or tissues and the body's antioxidant capacity [37].

The antioxidant levels of crustaceans change depending on environmental conditions. Wang et al. showed that under high salinity stress the antioxidant level of Chinese mitten crabs increased significantly [38]. In our study, we detected few significant differences in the antioxidant system of crabs between the two culture modes, indicating that polyculture would not cause oxidative stress damage to the crabs. For largemouth bass, the SOD activity in the EP group was always highest and the CAT activity in the EP group was significantly higher than that in the MM group at T2, which indicated that the antioxidant capacity of largemouth bass was higher under the polyculture mode. These results suggest that the polyculture of crabs and largemouth bass can activate the antioxidant system, increase the activity of GSH-Px, and act synergistically with SOD and CAT to reduce and block the damage caused by lipid peroxidation.

Because crustaceans lack an efficient adaptive immune system, ACP and AKP act as molecular effectors to protect them from pathogens and oxidative stress [39,40]. In the immune defense system, ACP and AKP directly regulate the transfer of phosphate groups and the metabolism of phosphate, which can accelerate the uptake and transport of substances in the body, form a hydrolase system, and destroy and remove foreign substances that invade the body, which is of great significance for maintaining the survival and growth of crabs and largemouth bass [41,42]. ACP is a marker enzyme of macrophage lysosomes and one of the most representative hydrolases in macrophages. As an immune indicator, AKP is a glycoprotein with a metallophosphatase structure. Its activity is a useful bioindicator of the physiological health of cellular membranes, cell growth, apoptosis,

cell migration, cellular metabolic status, hepatocyte function, and detoxification activity in hepatocytes [43,44].

In this study, the difference in ACP activity between the EM and EP culture modes in female crabs and between the MM and EP modes in largemouth bass at T1 may be part of an adaptation process in the pond. The AKP activity of female crabs in the EM group was higher than that in the EP group, but the difference was not significant. This may indicate that mixed culture with largemouth bass has little effect on the innate immunity of crabs. The AKP activities in male crabs in both groups at T4 were significantly higher than those of the other time points. The AKP activity of largemouth bass in the EP group was significantly higher than that in the MM group at both T1 and T2. Chen et al. previously showed when affected by salinity, the increase in ACP and AKP activities helps boost the non-specific immunity of aquatic organisms [45]. Our results show that the polyculture mode can improve the phosphatase activity of crabs and largemouth bass, which can help maintain their health.

Compared with Chinese mitten crabs collected at the beginning of the study, the proximate composition of the edible tissues did not change significantly over time. However, at the end of the experiment, the ash content in the dorsal muscle of largemouth bass in the EP group was significantly higher than that in the MM group. Ash is a measure of the mineral content of a food item [46], thus the mineral content in the dorsal muscle of largemouth bass in the EP group was higher than that in the MM group. This result may be due to the fish ingesting part of the crab feed and chilled fish in the polyculture mode.

Fatty acids are not only nutritionally important, but they also have a major impact on meat flavor because they influence the taste [47]. Moreover, essential fatty acids are an important basis for evaluating the nutritional quality of aquatic products [48]. SFAs mainly provide energy to animals [49]. In the present study, the content of $\sum$SFA in the edible tissues of female crabs in the EP group was higher than that in the EM group. This may be explained by the crabs' need for more reserved energy to cope with intense competition. The PUFA/SFA ratio is often used to evaluate the nutritional quality of meat, with a high ratio indicating good quality. The PUFA/SFA ratio of male crabs was higher than that of female crabs, indicating a higher nutritional quality of the edible tissues of male crabs. The content of PUFAs C18:2 and C20:2 in largemouth bass in the EP group were significantly higher than those in the MM group, but the content of C14:0 in the EP group was lower than that in the MM group. Fatty acid consumption and energy transfer occur in fish during exercise [50], and this may explain the differences in fatty acid content in the dorsal muscles of largemouth bass reared in the different culture modes.

Among the indicators of the quality of aquatic products, the ratio of EEAs to total hydrolyzed amino acids (E/T) is extremely important. According to the essential amino acid model of protein nutritional value proposed by the World Health Organization and the United Nations Food and Agriculture Organization in 1973, the EAA/(EAA + NEAA) value in an ideal protein should be about 40% [51]. The ratio in the muscle of crabs in the EM and EP groups in this study was close to this ideal value, which indicates that the amino acid nutritional value of crab muscle is rich. The hydrolyzed amino acids in the muscles of female and male crabs were all high in Glu, Leu, and Asp. Glu and Asp have a sour taste, and they are important prerequisite substances for the formation of umami substances [52]. Leu is slightly bitter and mainly plays a role in improving food flavor [53]. In largemouth bass, the content of hydrolyzed amino acids in the EP group was not significantly different from that in the MM group. From the perspective of hydrolyzed amino acids, the polyculture mode did not affect the quality of crab or largemouth bass amino acids.

Amino acids are the basic unit of protein composition and serve as an important nutritional index in the evaluation of the meat quality of aquatic products [54,55]. FAAs have different tastes, and the tastes are diverse but can be divided into three types: umami, sweet, and bitter [56]. Glu and Asp have an important impact on the umami taste of fish meat [57]. Gly and Ala provide a sweet taste and also reduce bitterness, remove unpleasant

tastes, and improve the overall taste [58]. For example, Ala was found to improve the flavor and taste of hake (*Eleginus gracilis*) [59]. In the present study, Ala, Arg, and Pro were the most abundant in FAAs in Chinese mitten crabs. The overall levels of Ala and FAAs in crabs in the EP group did not differ significantly compared with the EM group, indicating that the polyculture mode may not affect the flavor of FAAs in their edible tissues. The Arg content of female crabs was higher in the EP group than in the EM group, indicating that polyculture was favorable for the accumulation of Arg. The contents of Glu, Asp, and Met in largemouth bass in the EP group were lower than those in the MM group, and there was no significant difference in total FAAs between the EP and MM groups. From the perspective of FAAs, the flavor of largemouth bass may be lost to a certain extent in the crab–bass polyculture mode.

Flavor nucleotides have a synergistic effect on flavor taste when they coexist with FAAs [60]. For example, AMP can inhibit bitterness, and IMP and GMP can improve the umami taste [61,62]. In this study, both female and male crabs had high levels of AMP in both the EM and EP groups, and the IMP content of female crabs in the EP group was slightly higher than that in the EM group. The high content of AMP, IMP, Arg, Gly, Ala, and Glu have strong effects on crab meat flavor, and IMP nucleotides can act synergistically with Glu to enhance its perception [63]. Thus, we suggest that the polyculture mode may help increase the umami and sweet taste of the edible parts of female crabs. Interestingly, the largemouth bass dorsal muscle had higher GMP in the EP group than in the MM group. Given that GMP is known to produce umami taste, the umami of largemouth bass was improved by polyculture.

## 5. Conclusions

The results of this study show that the polyculture mode improved the size of Chinese mitten crabs, especially male crabs. It also increased the body mass and weight gain rate of largemouth bass and enhanced their oxidative stress capacity. Moreover, the nutritional value and flavor quality of the edible tissues in *E. sinensis* were improved to a certain extent under this polyculture mode, as indicated by the increased PUFA/SFA ratio and the increased content of Arg. This polyculture mode also improved the flavor quality of largemouth bass muscle to a certain extent. Our results provide a reference standard for different polyculture modes.

**Author Contributions:** Sample collection, data analysis, writing original draft, experiments, and editing: S.C.; sample collection: Y.S.; sample collection and data analysis: S.L.; sample collection, methodology, reviewing, and editing: Q.L.; methodology, J.G., Z.Z., Z.N., Z.T. and P.W.; conceptualization, methodology, reviewing, and editing: G.X. All authors have read and agreed to the published version of the manuscript.

**Funding:** This work was supported by the "JBGS" Project of Seed Industry Revitalization in Jiangsu Province [JBGS [2021]125] and the Central Public-interest Scientific Institution Basal Research Fund, CAFS (no. 2021XT0701).

**Institutional Review Board Statement:** The animal study protocol was approved by the Ethics Committee of Freshwater Fisheries Research Center of the Chinese Academy of Fishery Sciences (CAFS) (protocol code: HX200110001 and date of approval: 10 January 2020).

**Informed Consent Statement:** Not applicable.

**Data Availability Statement:** The datasets generated and/or analyzed for the current study are available from the corresponding author upon reasonable request.

**Conflicts of Interest:** There is no conflict of interest in the present study.

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
