# Peer review of "Polyculture Affects the Growth, Antioxidant Status, Nutrient Content, and Flavor of Chinese Mitten Crabs (Eriocheir sinensis) and Largemouth Bass (Micropterus salmoides)"

_fishes, doi:10.3390/fishes7060355_

Round 1

Reviewer 1 Report

I suggest a good English editing as some parts are not clear.

I have to admit that I am not a supporter of breeding alien species , in this case Micropterus salmoides. Breeding them is a loss of biodiversity - they should improve the breeding of authochthonous fish. However, the results and discussion are fine - even if they are reporting the presence of substances that at a first sight are not immediately clear why they are relevant.

I have written some remarks in the attached pdf.

Reviewer 2 Report

The authors have done an interesting work but the abstract does not reflect the work and has to be thoroughly revised as well as the specific objectives at the end of the introduction.The material and methods should be revised, a lot of information is missing, such as the type and composition of the diet provided, the quantity provided, the type of tanks used, how were they equipped, with aeration? without aeration? and how were the water exchanges performed, was it an open or closed system? There is also no detail on the equipment used to measure the water quality. Here are a few more points:

 Line 26 - AKP and ACP

Line 35 – insert the authority of the species Eriocheir sinensis. The authority should be used in the first time that you mentioned a scientific name

Line 41 – insert also the scientific name of the fish species and the authority

Lines 48-49 – The sentence that mentioned aquaponics is out of context, please remove and you can explore more the polyculture of crustaceans and fishes that is in line with your research.

Line 60 – revise the sentence

Line 60 to 64 – and the immune paramenters mentioned in the abstract? Revise the paragraph

Line 76 – size, form and volume of the tanks? And please insert the material of the tanks, mud tanks or concrete tanks?

Line 84 – Insert details of the diet, type, size, proximal composition…

Line 84 to 85 – revise the sentence

Line 91 – insert units and accuracy of “wet weights, carapace length and width”

Line 105-106 – insert the formulas

Line 162 – In the legends of tables and figures the scientific names should always be complete, please revise

Some sentences need to be revised; English has to be improved.

Round 2

Reviewer 1 Report

The following version is enhanced, and ready for publication.

I still keep my ethical concerns regarding the use of non autochthonous species.

Reviewer 2 Report

The authors have greatly improved the manuscript but there are still questions that have not been answered. For example: 

1 - The inclusion of the authority of each species has not been included, see for example WORMS to see what is accepted.

2 - The objectives still need major revision, for example it should not say that the objective is to test polyculture of the two species? The objective should be in accordance with the title and the methodology, it is not enough to say two groups. And they are no more than two groups plus control? in the experimental design the authors presented 3 groups, it is extremely confusing. For example the enzyme activity does not appear in the objectives or title, why was it performed? Please rewrite the objectives.

3 - Line 92 - They who?

4 - Line 86 how many control groups were used? Please insert the information.

5 - The proximal composition of the diet cannot be presented like this, it should be presented according to the correct values and not greater or equal. It should be noted that any trial is supposed to be replicable.

6 - The authors still do not indicate how many samples were used for the various analyses, how many samples per group were analyzed? Insert for the "n" for each one.

7 - The water quality should be presented with the values plus or minus the standard deviation, please revise.

8 - Lines 120-121 - insert the units in the formulas or outside them (g)

9 - English writing still needs revision.
